# Effect of Acid Whey in Combination with Sodium Ascorbate on Selected Parameters Related to Proteolysis in Uncured Dry-Fermented Sausages

Małgorzata Karwowska [1,*], Anna Kononiuk [2] and Dariusz M. Stasiak [1]

1 Department of Meat Technology and Food Quality, University of Life Sciences in Lublin, ul. Skromna 8, 20-704 Lublin, Poland
2 Institute of Animal Reproduction and Food Research of the Polish Academy of Sciences in Olsztyn, ul. Tuwima 10, 10-748 Olsztyn, Poland
* Correspondence: malgorzata.karwowska@up.lublin.pl

**Abstract:** The studies concern the effect of the addition of acid whey in combination with sodium ascorbate on selected parameters related to proteolysis in uncured dry-fermented sausages. Four sausage samples (with different additives: curing mixture (C); sea salt (S); sea salt and liquid acid whey (SAW); sea salt, liquid acid whey and sodium ascorbate (SAWA)) were taken at day 0, 7, 14 and 21 of ripening to assess basic physicochemical properties, thiobarbituric acid-reactive substances (TBARS), peptides content, antioxidant properties and biogenic amines content. It was indicated that the addition the acid whey caused greater acidification of fermented sausages during processing and significantly lower level of biogenic amines and lipid oxidation. At 21 days, the pH and water activity of sausages ranged from $4.74 \pm 0.01$ to $5.04 \pm 0.04$ and from $0.868 \pm 0.002$ to $0.875 \pm 0.003$, respectively. The synergistic effect of whey and sodium ascorbate was particularly noticeable in the reduction of lipid oxidation products and antioxidant activity of peptides. The results showed a beneficial effect of the addition of acid whey in combination with sodium ascorbate on the quality features of fermented sausage.

**Keywords:** fermented sausage; acid whey; ascorbic acid; peptides; biogenic amines

## 1. Introduction

Sausages are among the oldest man-made food products. Nowadays, many varieties of these foods are produced worldwide, which is related to with the availability of various types of raw meat, the climatic factors in different regions, the cultural and religious conditionings, and the processing methods passed on to future generations [1]. Among all the sausage varieties, the production of dry-fermented sausages is particularly important as these meat products are perceived by consumers as particularly valuable in terms of nutritional value [2]. The organoleptic characteristics of these products are the result of the many changes that occur with the raw materials and recipe ingredients thanks to the activity of the meat tissue enzymes and the microorganisms present. The processes that occur in fermentation, ripening and drying stages make changes in carbohydrates, proteins and lipids, which lead to gain the specific for fermented products values of the parameters including pH, color, taste, aroma and flavor [3]. Moreover, they contribute to increasing the presence of bioactive compounds such as bioactive peptides. Natural peptides extracted from meat and meat products are currently undergoing scientific research as biologically active compounds that exhibit various properties valuable for human health (including antioxidant properties, immunomodulatory effects and protection against oxidative stress) [4,5].

The above-mentioned content of peptides derived from fermented meat can be changed by modifying the processing conditions or the strains composing of the starter

culture [6]. The meat industry, also related to the production of raw fermented products, is under pressure from consumers and nutritionists to introduce other strategies to increase the nutritional value of processed meat. One such strategy is to remove harmful additives in line with the "clean label" trend [7]. The elimination of nitrite or nitrate used in the curing process [8] from the recipe of meat products increases the health benefits of processed meat. This is because, according to the literature, nitrogen compounds have negative health effects [9,10]. However, the elimination of nitrate/nitrite leads to some technological problems due to the multidirectional functions that these additives perform in meat products. Nitrates and nitrites play a role in inhibition the growth of bacteria (mainly *Clostridium botulinum*), limiting the process of oxidation and giving the typical reddish color and flavor of the processed meat [10]. As described by Honikel [11], the antioxidant properties are caused by nitrite being oxidized to nitrate by sequestering oxygen, which is not available for oxidizing fatty acids in this form. Similarly, the stable complexes arising between nitrite-derived compounds and heme-bond iron inhibit the release of free $Fe^{2+}$, which is the initiator of the reaction of lipid peroxidation [12]. Some researchers also provide other explanations about the antioxidant properties of nitrites [13]. The authors present a mechanism in which nitrite and dinitrogentrioxides react with unsaturated lipids, causing nitro-nitroso derivatives formation, which contributes to the inhibition of lipid peroxidative changes.

Given the multidirectional properties of nitrite/nitrates, many researchers are looking for potential alternatives to nitrite/nitrate salts that can be applied in the production of meat products [14,15]. Our previous studies were on acid whey application as an alternative to nitrite/nitrate salts in fermented meat products [14–19]. As indicated, the addition of acid whey in dry fermented sausage production improves its quality in terms of the heme iron content and fatty acid profile. Unfortunately, taking into account some of the properties of the products, acid whey turned out to not show the intended results. As indicated in the study on bioactive compounds in fermented sausages [20], the acid whey-caused application caused the decrease of peptides content, especially in fallow deer sausages. Looking for treatments to improve the effect of acid whey, we hypothesized that using acid whey in combination with sodium ascorbate will affect the processes occurring during the production of uncured dry fermented sausages. It has been indicated that the use of ascorbate salt in meat products causes the lowering of lipid oxidation [21] due to its ability to interact with iron ions, which are recognized as catalysts for the oxidation of lipids in meat products.

In this context, the aim of this study was to evaluate the effect of acid whey in combination with sodium ascorbate on basic physicochemical properties, thiobarbituric acid reactive substances (TBARS), peptides content, its antioxidant properties as well as biogenic amines content in fermented sausages during processing.

## 2. Materials and Methods

### 2.1. Raw Material

Raw material for production were beef leg and beef tallow from certificate organic farms, bought from the meat factory "Jasiolka" in Dukla. Fresh liquid acid whey was obtained from the dairy processing plant as a by-product of cottage cheese production. The pH and moisture content of liquid acid whey were, respectively, 4.74 ± 0.01 and 91.7 ± 0.13%. During production, noniodinated, and without anticaking agents, sea salt and glucose were used.

### 2.2. Dry-Fermented Sausage Production

The sausage manufacture consisted of four different formulations. Three of them were prepared adding sea salt (in these formulations, the addition of liquid acid whey and sodium ascorbate was also differentiated) and a fourth formulation control with sea salt and sodium nitrite.

For meats and tallows grinding, a Universal Machine KU2-3EK (Mesko-AGD, Skarżysko-Kamienna, Poland) was applied equipped with a 10-mm grinding plate. The formulation of dry fermented sausage assumed the use of meat and tallow in the ratio of 80:20. The raw meat and tallow were divided into four parts, each for separate treatments. After that, four treatments with different additives were prepared: C—with 2.8% curing mixture (99.5% sea salt and 0.5% sodium nitrite); S—with 2.8% sea salt; SAW—with 2.8% sea salt and 5% liquid acid whey; and SAWA—with 2.8% sea salt, 5% liquid acid whey and sodium ascorbate (0.05%). To each treatment, 0.6% of glucose was added. Water (5%) was added to the control sample to equalize the water level with the treatments to which liquid acid whey was added. Table 1 presents the formulation of dry fermented sausage. Fibrous casings (ø 65 mm; Viskase, Chicago, IL, USA) were used for stuffing. At least 8 kg of sausage samples of each formulation (the weight before the sausages were placed in the chamber) was prepared in each batch. The weight of one batter, before the sausages were placed in the chamber, was 0.8–1 kg. After each batter was weighed, the raw products were placed in the chamber in 16 °C with a relative humidity between 80% and 90% until for 21 days. The temperature and relative humidity were automatically regulated in the chamber. During this time, about $30\% \pm 3\%$ weight loss of sausages was achieved. Weight loss was checked by weighing all batters during maturation in the chamber with a PS1000.X2 scale (RAWAG, Radom, Poland).

**Table 1.** Formulation of dry fermented sausage.

| Ingredients (%) | C | S | SAW | SAWA |
|---|---|---|---|---|
| Sea salt | | + | + | + |
| Curing mixture | + | | | |
| Glucose | + | + | + | + |
| Liquid acid whey | | | + | + |
| Water | + | | | |
| Ascorbic acid | | | | + |

C—with curing mixture; S—with sea salt; SAW—with sea salt and liquid acid whey; SAWA—with sea salt, liquid acid whey and sodium ascorbate.

Total nitrogen content (TN), non-protein nitrogen content (NPN), pH, heme iron content and water activity ($a_w$) were determined in the stuffing (0 day) and in the 7th, 14th and 21st day of manufacturing process. Peptide content and their antioxidant activity (ABTS) were measured once at the end of the manufacturing process. The treatments were replicated twice by producing two different batches.

*2.3. pH and Water Activity ($a_w$) and Thiobarbituric Acid Reactive Substances (TBARS) Measurements*

The pH of the homogenates (sample homogenized with distilled water) was measured with a digital pH meter CPC-501 (Elmetron, Zabrze, Poland) equipped with a temperature sensor and pH electrode (ERH-111, Hydromet, Gliwice, Poland). Before analysis, the pH meter was calibrated with buffer solutions at pH 4.0, 7.0 and 9.0. The water activity ($a_w$) was measured at 20 °C using a water activity analyzer (Novasina AG, Lachen, Switzerland) calibrated with Novasina SAL-T humidity standards (33%, 75%, 84% and 90% relative humidity). The amount of thiobarbituric acid reactive substances (TBARS) was measured based on the procedure described by Pikul et al. [22] to assess the degree of lipid oxidation in samples of meat products. The TBARS values were expressed as mg MDA/kg sample.

### 2.4. Determination of Total Nitrogen Content (TN) and Non-Protein Nitrogen (NPN)

Total nitrogen content (expressed as a mass percent composition) was measured by the Kjeldahl method. NPN (expressed as a mass percent composition) was measured according to the method of Careri et al. [23] with slight modifications.

### 2.5. Total Pigments (TP) and Heme Iron (HI) Content

The TP and HI contents were measured according to the method described by Karwowska and Dolatowski [24]. UV-visible spectrophotometer (Nicolet Evolution 300, Thermo Electron Corp., Waltham, MA, USA) was used to measure the absorbance of the samples. The total pigments were calculated as hematin and expressed as mg kg$^{-1}$. The heme iron was calculated based on total pigment content.

### 2.6. Determination of Peptide Content and Its Antioxidant Activity

The peptide extraction was performed according to the method of Zhu et al. [25] with slight modifications described in our previous study [20]; 25 g of sample was homogenized with 100 mL of 0.01 M HCl with cooling ice. The homogenate was centrifuged and then filtrate through glass wool. Then, 25 mL of supernatant was added to 75 mL of frozen ethanol. The mixture was kept in 4 °C by night and then centrifuged. The supernatant was collected and concentrate in an evaporator. The concentrated extract was dissolved in 0.01 M HCl and filtered through a 0.45 μm nylon membrane filter. The peptides content was determined using o-phthaldialdehyde (OPA) assay [25]. The extraction was performed in triplicate.

The antioxidant activity against ABTS$^{\bullet+}$ (2,2-azinobis(3-ethyl-benzothiazoline)-6-sulfonate) radical cation of the extracted peptides was determined based on the method described by Re et al. [26]. Trolox was used as an oxidant standard. Briefly, ABTS$^{\bullet+}$ radical cation was produced by reacting ABTS stock solution (7 Mm) with potassium persulfate and allowing the mixture to stand in the dark at room temperature for 12–16 h before use. The stock solution of ABTS was diluted with PBS (pH 7.4) to an absorbance of 0.70 ($\pm$0.02) at 734 nm, then 1.9 mL of diluted ABTS$^{\bullet+}$ solution ($A_{734nm} = 0.700 \pm 0.020$) and 100 μL of extracted peptide (concentration 1 mg/mL) or Trolox standards (final concentration 0–7.5 mg mL$^{-1}$) in PBS were mixed. The antioxidant activity of peptide concentration 1 mg mL$^{-1}$ was expressed as equivalents of mg Trolo $\times$ 100 mL$^{-1}$.

### 2.7. Biogenic Amines (BAs) Determination

Biogenic amines (BAs) were determinate in experimental sausages samples at the end of production using an AAA 500 amino acid analyzer (Ingos, Praha, Czech Republic), equipped with an Ostion LG AAA8 ion-exchange column (3.6 $\times$ 100 μm, 8 μm). BAs were extracted with 10% trichloroacetic acid according to previously described procedure [27]. The content of the BAs was measured with a reference to the amine standards (histamine, tyramine, putrescine, cadaverine, spermidine, agmatine, spermine) (Ingos, Prague, Czech Republic). The BAs concentration was expressed as mg kg$^{-1}$ of dry fermented sausages.

### 2.8. Statistical Analysis

The experiment was replicated three times by producing three different batches on separate days. Data were analyzed using Statistica TM v.13.3 software (1984) from Statsoft (StatSoft Inc., Tulsa, OK, USA). Post-hoc comparations were identified using Tukey's honest significant difference (HSD) test. Data were expressed as the mean $\pm$ standard deviation (SD). The outliers have been determined using the interquartile range. Variables determined at the end of processing such as content of peptide and their antioxidant activity, biogenic amines content were analyzed using one-way ANOVA. Other variables, which were determined during processing, were analyzed using factorial ANOVA to evaluate the effect of processing time and treatments. The correctness of the use iof analysis of variance in both cases were confirmed by Leavene Test for homogeneity. Post-hoc comparison was specified based on Tukey's test. All differences were significant at $p \leq 0.05$. To

evaluate the correlations between parameters investigated during processing, principal component analysis (PCA) was applied. The correctness of using PCA was checked using the Kaiser–Meyer–Olkin (K M O) test for sampling adequacy and Bartlett's test of sphericity. The number of principal components was determined based on percentages of variance explained using Kaiser's criterion and Cattell's test.

## 3. Results

### 3.1. pH, Water Activity, TBARS

Table 2 presents the pH, water activity and TBARS values of the dry fermented sausages. At the beginning of the experiment (day 0), the pHs differed significantly between treatments and ranged from $5.98 \pm 0.02$ to $6.04 \pm 0.02$. The pH of the sausage sample with sea salt (S) was significantly ($p \leq 0.05$) higher compared to the sample with curing mixture (C) and sausage sample with sea salt, liquid acid whey and sodium ascorbate (SAWA). During the manufacturing process, significant differences between the samples C and S were also detected. However, data regarding processing time showed that the pH decreased significantly ($p \leq 0.05$) in all samples during ripening. The decrease in pH was significant after every 7 days of manufacturing process in samples C, S and SAW. In the case of the sample with ascorbic acid (SAWA), a decrease in pH value in the stuff and after 7 days was not significantly different ($p \leq 0.05$), but in next period time the changes were also significant. The addition of acid whey caused the greater acidification of fermented sausages at 7, 14 and 21 days compared to cured (C) and salted (S) sausages.

**Table 2.** pH, water activity and TBARS of dry fermented sausages during processing (mean ± SD).

| Parameter | Treatments | Day | | | |
|---|---|---|---|---|---|
| | | 0 | 7 | 14 | 21 |
| pH | C | 5.98 [Ab] ± 0.02 | 5.64 [Ba] ± 0.01 | 5.50 [Ca] ± 0.03 | 5.04 [Da] ± 0.04 |
| | S | 6.04 [Aa] ± 0.02 | 5.56 [Bb] ± 0.03 | 5.44 [Cb] ± 0.02 | 4.97 [Cb] ± 0.02 |
| | SAW | 5.99 [Aab] ± 0.01 | 5.40 [Bc] ± 0.01 | 5.19 [Cd] ± 0.02 | 4.74 [Cd] ± 0.01 |
| | SAWA | 5.98 [Ab] ± 0.01 | 5.56 [Bb] ± 0.01 | 5.32 [Cc] ± 0.08 | 4.86 [Cc] ± 0.06 |
| $a_w$ | C | 0.964 [Aa] ± 0.003 | 0.921 [Ba] ± 0.001 | 0.911 [Ca] ± 0.001 | 0.875 [Da] ± 0.003 |
| | S | 0.967 [Aa] ± 0.001 | 0.919 [Ba] ± 0.005 | 0.909 [Cab] ± 0.002 | 0.875 [Da] ± 0.002 |
| | SAW | 0.968 [Aa] ± 0.001 | 0.923 [Ba] ± 0.001 | 0.904 [Cbc] ± 0.01 | 0.868 [Db] ± 0.002 |
| | SAWA | 0.967 [Aa] ± 0.001 | 0.918 [Ba] ± 0.01 | 0.899 [Cc] ± 0.003 | 0.867 [Db] ± 0.002 |
| TBARS (mg kg⁻¹) | C | 2.10 [Aa] ± 0.13 | 2.76 [Aa] ± 0.11 | 2.45 [Aa] ± 0.14 | 2.53 [Aa] ± 0.13 |
| | S | 2.69 [Aa] ± 0.05 | 2.42 [Aa] ± 0.49 | 2.48 [Aa] ± 0.10 | 2.44 [Aa] ± 0.18 |
| | SAW | 2.49 [Aa] ± 0.03 | 2.69 [Aa] ± 0.20 | 2.54 [Aa] ± 0.11 | 2.33 [Aa] ± 0.11 |
| | SAWA | 1.47 [Ab] ± 0.06 | 1.13 [Ab] ± 0.29 | 1.29 [Ab] ± 0.06 | 1.34 [Ab] ± 0.10 |

C—with curing mixture; S—with sea salt; SAW—with sea salt and liquid acid whey; SAWA—with sea salt, liquid acid whey and sodium ascorbate. Means marked with the same lowercase letters [a–d] in the same column are not statistically different from each other ($p \leq 0.05$). Means marked with the same capital letter [A–D] in the same line (within the same variant of variable in different time) are not statistically different from each other ($p \leq 0.05$).

During the manufacturing, water activity ($a_w$) decreased significantly in all samples (Table 2). The highest changes in $a_w$ values were observed between stuff and after 7 days of manufacturing as well as between 14th and 21st day of processing. At 21 day water activity ranged from $0.867 \pm 0.002$ in SAW sample to $0.875 \pm 0.003$ in C and S samples. The differences in water activity in the samples with acid whey (SAW, SAWA) and other samples (C, S) were statistically significant at the end of production. Regarding the lipid oxidation, during the production of dry fermented sausages, no significant influence of the processing time on the TBARS value was observed (Table 2). However, acid whey and sodium ascorbate's combined addition resulted in significant changes of the TBARS value compared to acid whey separate addition or the treatment C and S. At 0, 7,14 and 21 days, samples with acid whey and ascorbic acid (SAWA) were characterized by significantly lower TBARS values compared to other samples.

### 3.2. Content of Total and Non-Protein Nitrogen, Proteolysis Index, Total Content of Pigments and Heme Iron of Dry Fermented Sausages during Processing

Total nitrogen and non-protein nitrogen content during dry-fermented sausage production increased significantly ($p \leq 0.05$), as shown in Table 3. In the case of total nitrogen, no significant differences between samples at the same process time were observed. In the case of non-protein nitrogen, a significant increase with processing time was noted for all samples. Moreover, the elimination of sodium nitrite and the addition of acid whey as well as acid whey in combination with sodium ascorbate did not have a significant effect on the discussed parameters. Proteolysis index (PI) was calculated as a ratio between NPN and TN contents; it shows intensity of proteolysis. Changes in PI during processing were shown in Table 3. During the processing, PI increased for all samples. The biggest change in PI, which suggests that the intensity of proteolysis rapidly increases, was observed in samples with acid whey addition (SAW, SAWA) between the stuff sample and sample in the seventh day of manufacturing. Among the factors included in ANOVA, both time and treatment showed a significant effect of the total heme pigments and heme iron content of dry-fermented sausages. As expected, ripening time resulted in an increase in the content of total heme pigments and heme iron due to lowering the water content as a result of drying. Between treatments, statistically significant differences occurred only on day 14. The sample with acid whey in combination with sodium ascorbate was characterized by significantly higher total heme pigments and heme iron content compared to cured and salted samples (C, S).

**Table 3.** Content of total and non-protein nitrogen, proteolysis index, total content of pigments and heme iron of dry fermented sausages during processing (mean $\pm$ SD).

| Parameter | Treatments | Day | | | |
|---|---|---|---|---|---|
| | | 0 | 7 | 14 | 21 |
| Total nitrogen (mg g$^{-1}$) | C | 2.85 $^{Ca}$ $\pm$ 0.06 | 3.38 $^{Ba}$ $\pm$ 0.10 | 3.69 $^{ABab}$ $\pm$ 0.19 | 4.24 $^{Aa}$ $\pm$ 0.22 |
| | S | 2.79 $^{Ca}$ $\pm$ 0.19 | 3.34 $^{Ba}$ $\pm$ 0.18 | 3.40 $^{Bb}$ $\pm$ 0.15 | 4.11 $^{Aa}$ $\pm$ 0.13 |
| | SAW | 2.95 $^{Ba}$ $\pm$ 0.13 | 3.19 $^{Ba}$ $\pm$ 0.11 | 3.99 $^{Aa}$ $\pm$ 0.13 | 4.22 $^{Aa}$ $\pm$ 0.16 |
| | SAWA | 2.95 $^{Ba}$ $\pm$ 0.21 | 3.21 $^{Ba}$ $\pm$ 0.11 | 3.98 $^{Aa}$ $\pm$ 0.14 | 4.13 $^{Aa}$ $\pm$ 0.13 |
| Non protein nitrogen (mg g$^{-1}$) | C | 0.19 $^{Ca}$ $\pm$ 0.00 | 0.25 $^{Ba}$ $\pm$ 0.01 | 0.28 $^{Ba}$ $\pm$ 0.03 | 0.34 $^{Aa}$ $\pm$ 0.01 |
| | S | 0.17 $^{Ca}$ $\pm$ 0.00 | 0.23 $^{Ba}$ $\pm$ 0.01 | 0.26 $^{Ba}$ $\pm$ 0.01 | 0.34 $^{Aa}$ $\pm$ 0.02 |
| | SAW | 0.16 $^{Da}$ $\pm$ 0.00 | 0.24 $^{Ca}$ $\pm$ 0.01 | 0.29 $^{Ba}$ $\pm$ 0.01 | 0.34 $^{Aa}$ $\pm$ 0.00 |
| | SAWA | 0.17 $^{Ca}$ $\pm$ 0.00 | 0.24 $^{Ba}$ $\pm$ 0.00 | 0.31 $^{Aa}$ $\pm$ 0.02 | 0.32 $^{Aa}$ $\pm$ 0.01 |
| Proteolysis index | C | 6.62 $^{Ba}$ $\pm$ 0.15 | 7.25 $^{ABa}$ $\pm$ 0.31 | 7.57 $^{ABa}$ $\pm$ 0.62 | 7.99 $^{Aa}$ $\pm$ 0.34 |
| | S | 6.10 $^{Ba}$ $\pm$ 0.62 | 7.00 $^{ABa}$ $\pm$ 0.61 | 7.53 $^{Aa}$ $\pm$ 0.47 | 8.31 $^{Aa}$ $\pm$ 0.71 |
| | SAW | 5.54 $^{Ba}$ $\pm$ 0.41 | 7.41 $^{Aa}$ $\pm$ 0.37 | 7.26 $^{Aa}$ $\pm$ 0.33 | 7.97 $^{Aa}$ $\pm$ 0.21 |
| | SAWA | 5.63 $^{Ba}$ $\pm$ 0.46 | 7.45 $^{Aa}$ $\pm$ 0.26 | 7.71 $^{Aa}$ $\pm$ 0.38 | 7.75 $^{Aa}$ $\pm$ 0.23 |
| Total heme pigments (mg kg$^{-1}$) | C | 206.72 $^{Ba}$ $\pm$ 19.04 | 301.69 $^{Aa}$ $\pm$ 16.88 | 299.20 $^{Ab}$ $\pm$ 5.81 | 290.59 $^{Aa}$ $\pm$ 0.39 |
| | S | 196.23 $^{Ba}$ $\pm$ 20.43 | 276.53 $^{Aa}$ $\pm$ 13.69 | 304.41 $^{Ab}$ $\pm$ 10.63 | 315.52 $^{Aa}$ $\pm$ 6.56 |
| | SAW | 224.17 $^{Ba}$ $\pm$ 23.48 | 300.11 $^{Aa}$ $\pm$ 10.84 | 312.35 $^{Aab}$ $\pm$ 6.17 | 328.21 $^{Aa}$ $\pm$ 1.04 |
| | SAWA | 206.95 $^{Ca}$ $\pm$ 19.24 | 301.92 $^{Ba}$ $\pm$ 10.79 | 352.47 $^{Aa}$ $\pm$ 6.46 | 318.24 $^{ABa}$ $\pm$ 12.03 |
| Heme iron (mg kg$^{-1}$) | C | 18.23 $^{Ba}$ $\pm$ 1.68 | 26.61 $^{Aa}$ $\pm$ 1.49 | 26.39 $^{Ab}$ $\pm$ 0.51 | 25.63 $^{Aa}$ $\pm$ 0.03 |
| | S | 17.31 $^{Ba}$ $\pm$ 1.80 | 24.39 $^{Aa}$ $\pm$ 1.21 | 26.85 $^{Ab}$ $\pm$ 0.94 | 27.83 $^{Aa}$ $\pm$ 0.58 |
| | SAW | 19.77 $^{Ba}$ $\pm$ 2.07 | 26.74 $^{Aa}$ $\pm$ 0.96 | 27.55 $^{Aab}$ $\pm$ 0.54 | 28.95 $^{Aa}$ $\pm$ 0.09 |
| | SAWA | 18.25 $^{Ca}$ $\pm$ 1.70 | 26.63 $^{Ba}$ $\pm$ 0.95 | 31.09 $^{Aa}$ $\pm$ 0.57 | 28.07 $^{ABa}$ $\pm$ 1.06 |

C—with curing mixture; S—with sea salt; SAW—with sea salt and liquid acid whey; SAWA—with sea salt, liquid acid whey and sodium ascorbate. Means marked with the same lowercase letters [a,b] in the same column are not statistically different from each other ($p \leq 0.05$). Means marked with the same capital letter [A–D] in the same line (within the same variant of variable in different time) are not statistically different from each other ($p \leq 0.05$).

Principal component analysis (PCA) was made to determine which of the investigated during processing parameters have the greatest effect on the principal components. The correctness of using PCA analysis was analyzed using the KMO index and Bartlett's test of sphericity, which results were shown in Table 4. The KMO index (higher than 0.5) as well as results of Bartlett's test confirmed the correctness of application PCA analysis in this case. The results of PCA shown in Table 5 revealed that the first two principal components explained 88.94% of the cumulative variance, with 76.34% of total variance explained by the first principal component (PC1). The plane of the first two principal components is shown in the Figure 1. The highly positive loading of pH and water activity count in PC1 indicated the high dependence of these two variables during the experiment. A similar arrangement of variables with high negative loading (Figure 1) in PC1 has shown a significant effect between parameters related to proteolysis (TN, NPN, PI) as well as total pigments and heme iron content. Second principal component explained 12.60% of cumulative variance and is correlated highly only with TBARS.

**Table 4.** Results of KMO and Bartlett's tests calculated for parameters investigated during processing of sausages.

| Test | Result |
| --- | --- |
| KMO index | 0.75 |
| Bartlett's test of sphericity | |
| Approximate $\chi^2$ | 791.92 |
| Degrees of freedom | 28 |
| Significance | <0.000001 |

**Table 5.** Correlation of variables to factors in principal component analysis based on factor loadings.

| Variable | PC1 (76.34%) | PC2 (12.60%) |
| --- | --- | --- |
| pH | 0.954 | 0.067 |
| aw | 0.976 | 0.042 |
| Proteolysis index (PI) | −0.870 | −0.036 |
| Total nitrogen (TN) | −0.910 | −0.074 |
| Non protein nitrogen (NPN) | −0.974 | −0.064 |
| Total pigments (TP) | −0.922 | 0.090 |
| Heme iron (HI) | −0.922 | 0.090 |
| TBARS | 0.107 | −0.988 |

The content and antioxidant activity of peptides isolated from dry fermented sausages at the end of manufacturing (21 day) is shown in Table 6. The sausage samples were characterized by the mean peptide content from $9.69 \pm 1.19$ to $11.70 \pm 0.49$ mg g$^{-1}$ of product. The peptide content of salted sample (S) and sample with liquid acid whey and sodium ascorbate (SAWA) was significantly lower compared to the cured sample with sodium nitrite (C). However, the antioxidant activity of peptides was independent of their content because the sample SAWA with the lowest peptide content showed the highest antioxidant activity, which was significantly higher compared to the C and S sausage samples. The antioxidant activity of peptide isolated from dry fermented sausage with acid whey addition (SAW) did not differ significantly from the SAWA sample.

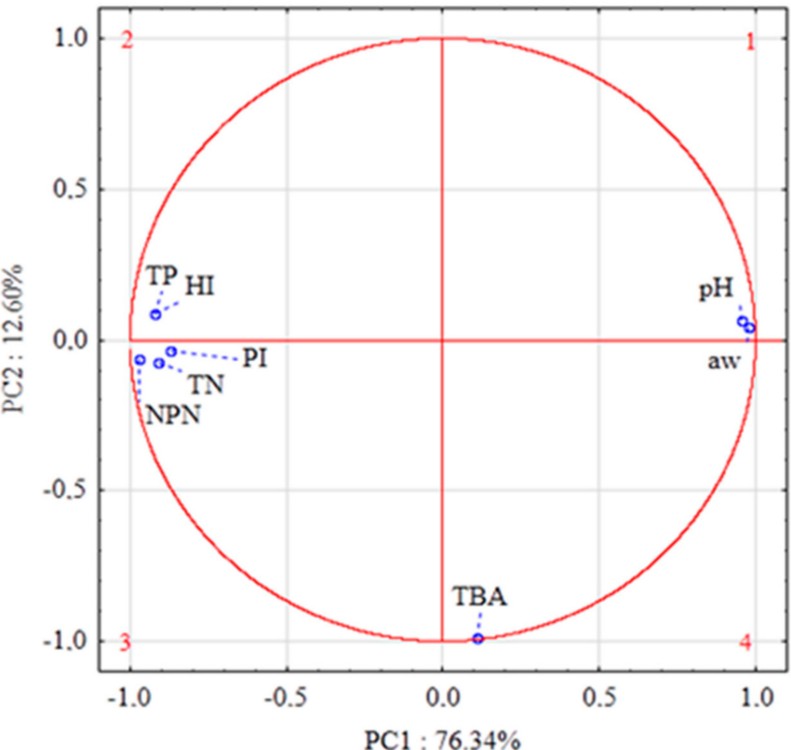

**Figure 1.** Loadings for the two principal components. TP—total pigment content, HI—heme iron content, PI—proteolysis index, TN—total nitrogen content, NPN—non protein nitrogen.

**Table 6.** Content and antioxidant activity of peptides isolated from dry fermented sausages at the end of manufacturing (21 day) (mean ± SD).

| Parameter | Treatments | Peptide |
|---|---|---|
| Peptide content (mg g$^{-1}$ of product) | C | 13.49 [a] ± 0.75 |
| | S | 11.05 [bc] ± 0.90 |
| | SAW | 11.70 [ab] ± 0.49 |
| | SAWA | 9.69 [c] ± 1.19 |
| Antioxidant activity of peptides (eq. mg Trolox 100 mL$^{-1}$) | C | 1.56 [b] ± 0.25 |
| | S | 1.84 [b] ± 0.35 |
| | SAW | 2.05 [ab] ± 0.06 |
| | SAWA | 2.49 [a] ± 0.32 |

C—with curing mixture; S—with sea salt; SAW—with sea salt and liquid acid whey; SAWA—with sea salt, liquid acid whey and sodium ascorbate. Means with the same lowercase letter [a–c] do not differ significantly within the variable ($p \leq 0.05$).

Table 7 reports the amount of the identified amines at the end of the manufacturing of dry-fermented sausages. Four biogenic amines were identified in the experimental sausages (tyramine, putrescine, cadaverine and spermine). Histamine, spermidine and agmantine were not detected. The detection limits of the method were 100 pmol for histamine and agmantine and 50 pmol for spermidine. The most abundant biogenic amine present in all sausages were the tyramine. The lowest tyramine content was found in the sausage sample with the addition of acid whey and sodium ascorbate, followed by the sample SAW with acid whey, then salted sample S. The highest concentration of this BA, twice as high as compared to the SAWA sample, was achieved for cured sample C with the addition of sodium nitrite. All sausage samples showed similar amounts of spermine, ranging from 32.65 ± 2.07 to 40.70 ± 10.47 mg kg$^{-1}$.

**Table 7.** Biogenic amines content in dry fermented sausages at the end of manufacturing (21 days) (mean ± SD).

| Biogenic Amine | Treatments | The Amount of Biogenic Amine (mg kg$^{-1}$) |
|---|---|---|
| Tyramine | C | 237.50 [a] ± 0.71 |
| | S | 212.00 [b] ± 2.83 |
| | SAW | 155.50 [c] ± 26.16 |
| | SAWA | 120.50 [d] ± 9.19 |
| Putrescine | C | 11.85 [b] ± 7.14 |
| | S | 38.60 [a] ± 20.36 |
| | SAW | 1.90 [b] ± 2.12 |
| | SAWA | 5.15 [b] ± 5.87 |
| Cadaverine | C | 98.10 [a] ± 23.90 |
| | S | 116.25 [a] ± 49.14 |
| | SAW | 51.90 [b] ± 19.66 |
| | SAWA | 69.55 [ab] ± 20.15 |
| Spermine | C | 39.60 [a] ± 1.27 |
| | S | 40.70 [a] ± 10.47 |
| | SAW | 32.65 [a] ± 2.07 |
| | SAWA | 35.15 [a] ± 2.62 |

C—with curing mixture; S—with sea salt; SAW—with sea salt and liquid acid whey; SAWA—with sea salt, liquid acid whey and sodium ascorbate. Means with the same lowercase letter [a–d] do not differ significantly within the variable ($p \leq 0.05$).

## 4. Discussion

During the production of fermented sausages, key stages (fermentation, ripening and drying) take place that modify the main meat components (carbohydrates, proteins and lipids) and thus determine the specific physicochemical parameters (pH, water activity) [3]. The acidification and water activity of fermented meat product are very important as these parameters mainly affect chemical and enzymatic reaction activity and growth of microorganisms during manufacturing [28]. The production of organic acids (mainly lactic acid), hydrogen peroxide and bacteriocins from microbial activity (naturally occurring or starter cultures) plays a crucial role in preventing the growth of spoilage microorganisms and food-borne pathogens in fermented sausages [29]. In the current study, the dry fermented sausages had an initial pH ranging from 5.98 ± 0.02 to 6.04 ± 0.02, which decreased to values ranging from 4.74 ± 0.01 to 5.04 ± 0.04 after 21 days of production. Such low pH values (<5.0) are an important factor inhibiting the growth of pathogenic microorganisms [30]. The greatest decrease in pH, by more than 0.5 units, occurred after seven days of production. The addition of the acid whey caused a greater acidification of fermented sausages at 7, 14 and 21 days compared to cured (C) and salted (S) sausages. Lower pH values of samples with the addition of acid whey mainly result from its composition. The addition of acid whey enables the introduction of lactic acid bacteria into the stuffing, which affects the acidification of the environment [31]. A similar low pH was obtained in the studies by Karwowska and Dolatowski [16], in which nitrite-/nitrate-free fermented sausage made from deer meat presented values for the pH from 4.47 to 4.59. In these studies, it was shown that the inclusion of acid whey to the sausage together with freeze-dried cranberries significantly decreased the pH values.

The effect of the addition of whey was also observed in the case of the results of the water activity of fermented sausages. The experimental sausage samples with the addition of acid whey and acid whey in combination with sodium ascorbate were characterized

by significantly lower water activity compared to the other tests at the end of processing. However, the water activity of experimental sausages, regardless of the additive used in the production process, was lower than the minimum necessary for the development of most pathogenic microorganisms, such as: *Pseudomonas* (aw > 0.97), *Clostridium botulinum* (aw > 0.93 ÷ 0.96), *Salmonella* (aw > 0.94) and *Listeria monocytogenes* (aw > 0.92) [30]. In our previous studies on the effect of the addition of acid whey [18], similarly low water activity values of fermented beef sausages were obtained (<0.900), which resulted in their classification to the class aw, which indicates unfavorable conditions preventing the growth of the most popular bacteria in meat products such as *Listeria monocytogenes*, *Clostridium botulinum* and *Escherichia coli*.

Parameters such as pH and water activity are also crucial in many of the processes involved in the production of fermented sausages. These properties are related to the proteolysis process that takes place during the production of fermented meat products. As a result of these changes, peptides and amino acids are formed from muscle proteins. The enzymes that participate in these transformations include cathepsins B, D, H and L and, to a lesser extent, calpain and exopeptidases (peptidases and aminopeptidases) [32]. Decrease of water activity limits the activity of muscle enzymes. According to data published by Toldra [33], a decrease of water activity under 0.9 strongly affects the activity of cathepsin and aminopeptidases and slightly decreases the calpain activity. In our research in stuff samples (0 day), the water activity was about 0.96, which suggests that the relative activity of enzymes could be above 70%. At the end of the process, water activity decreased to 0.867–0.875, and according to Toldra [33], the activity of enzymes was much less intensive. The higher changes in relative activity of cathepsin and aminopeptidases were observed when water activity decreased from 1 to 0.9 (relative activity of enzymes in some cases above 60% decreased) in relation to our results that showed that, in the first seven days of processing, the activity of enzymes was the most intense. Moreover, pH value at the beginning of the process (0 days) suggest that the maximal activity reached cathepsin B and L, which have shown good ability to degrade myofibrilar protein and are responsible for the degradation of myosin heavy chain, actin, titin and nebulin. After one week, the pH value inactivated activity of calpain I and II (pH = 5.5); however, that was the optimal pH for activity of dipeptydylpeptidases (DPP) I and II. A further decrease in pH favors activity of cathepsin D and tripeptidases I. In this context, in the current study, the proteolysis index (PI) was an indicator of the intensity of proteolysis increased for all samples during the processing. The PI values in the tested sausages after the end of production were similar to the values published by other authors [17,34,35]. The highest change in PI, which suggests that the intensity of proteolysis rapidly increase, was observed in samples between the stuff sample and sample in the seventh day of manufacturing. Similar trends were observed by Safa et al. [36] who, during the production of raw-ripening products, noted the highest increase in the proteolysis index after the first week of the ripening process. As in previous studies [34], no effect of acid whey supplements on the value of the proteolysis index was observed. Similarly, the use of sodium ascorbate did not significantly affect the intensity of proteolysis during the production of experimental fermented sausages.

The effect of proteolysis is the formation of peptides that were assessed in this study. The experimental sausages were characterized by the peptide content from $9.69 \pm 1.19$ to $11.70 \pm 0.49$ mg g$^{-1}$ of product. As in previous studies [20], the addition of acid whey did not increase the peptide content in fermented sausages, as expected. Moreover, the use of sodium ascorbate in combination with whey resulted in the lowest peptide content. However, the antioxidant activity of peptides was independent of their content because the sample with the lowest peptide content (with acid whey and sodium ascorbate) showed the highest antioxidant activity, significantly higher compared to the cured sausage sample. Moreover, the antioxidant activity of peptides isolated from dry fermented sausage with acid whey addition (SAW) did not differ significantly from the sausage sample produced with acid whey and sodium ascorbate. As shown by available scientific research, molecular weights and the amino acid sequence play a vital role in determining the antioxidant

activity of the peptides [37]. According to previous reports on peptides derived from other sources [38–40], smaller molecular weights of crude peptides tend to have higher antioxidant activity. Specific amino acid sequences of peptides are also important for their properties [41]. The amino sequence of Pro-His-His exhibited higher antioxidant capacity in the linoleic acid system than that of other synthetic peptides. Similarly, tripeptides with Trp and Tyr residue at the C-terminus could exert strong radical scavenging activities. Moreover, it was indicated that more than half of the peptide sequences derived from animal proteins with strong scavenging radical capacity contain hydrophobic amino acids.

The antioxidant properties of peptides are crucial for meat products' quality, as oxidative changes related to protein and lipids take place during the production of dry fermented sausages. These properties help to improve the shelf life of the products [42,43]. In the present study, TBARS was used to evaluate the degree of lipid oxidation of fermented sausages. As expected, acid whey and sodium ascorbate's combined addition resulted in significantly decreasing the level of secondary products of lipid oxidation. It has been shown that the application of ascorbate salt in meat products inhibits lipid oxidation [21,43] as this compound is involved in redox reactions. Ascorbate acts as an electron donor, and its oxidized form (dehydroascorbic acid) is relatively uncreative and therefore terminates the propagation of free radical reaction. Its ability to react with iron ions and thus elimination of the negative catalytic effect of iron ions is also known [21]. The protective properties of the additives used (acid whey in combination with sodium ascorbate) were also observed in the content of heme pigments and heme iron on the 14th day of production. During the storage and processing of meat, hemin could dissociate from the globin, and heme oxygenase and reactive oxygen species could cause heme destruction, which subsequently releases iron from the heme ring [44]. The consequence of this phenomenon is the reduction of heme iron and the formation of a form of "free" iron. This situation has been observed in current research during the production of raw fermented sausages. The sodium ascorbate used inhibited the destructive processes of myoglobin only until the 14th day of production. On the 21st day of production, all samples were characterized by a similar content of heme iron.

Fermented meat products are one of the most common dietary sources of biogenic amines (BAs). Although BAs can be found in a wide range of meat products; during fermented meat products production, especially suitable environmental conditions take place that favor the accumulation of these compounds due to the activity of microorganisms showing decarboxylase enzymes activity [21,45]. In this context, they are important as an index for product quality, but also for their impact on public health [44]. The most common BAs occurring in fermented foods are tyramine, putrescine and cadaverine [46,47]. In the current study, in addition to the indicated amines, small amounts of spermine were also detected. The use of acid whey and sodium ascorbate significantly reduced the content of biogenic amines in fermented sausages, in particular tyramine, which was detected in the highest amount. According to Schirone et al. [45], tyramine is the most represented amine in fermented meat products, and this has been confirmed also in this study. The amount of tyramine in fermented sausage sample with the addition of acid whey in combination with sodium ascorbate are similar to those reported by EFSA [48], which indicated a tyramine mean concentration of 136 mg/kg in 400 European salami samples. Similar results were obtained by other researchers who found that tyramine was the most abundant bioactive amine found in dry fermented sausages of Spanish retail [49] and salamis from Southern Italy [50]. Histamine was not found in the experimental fermented sausages. This is very beneficial as histamine is considered the most dangerous for human health because it can cause various adverse effects known as "histamine poisoning" [51]. Acidity is an important factor influencing the formation of amines, as the activity of amino acid decarboxylases is stronger in acidic conditions, with pH 4.0–5.5 [52]. In the current research, samples characterized by a lower pH were characterized by a lower content of biogenic amines, which is probably related to the lower presence of microorganisms showing decarboxylase enzymes activity. As related by Durak-Dados et al. [52], *Enterobacteriaceae* isolated from sausages

are generally considered microorganisms with high decarboxylase activity, particularly in relation to the production of cadaverine and putrescine. It can therefore be assumed that the samples with the addition of acid whey and acid whey in combination with sodium ascorbate are characterized by a lower content of *Enterobacteriaceae* showing activity of decarboxylase enzymes.

## 5. Conclusions

The incorporation of material rich in bioactive compounds, such as acid whey, in the preparation of fermented sausages is one of the novel strategies to develop meat products without nitrogen compounds according to the "clean label" trend. The obtained results indicate a positive effect of the addition of acid whey in combination with sodium ascorbate on the quality features of fermented sausage. Fermented meat products in which these additives were used showed the lowest level of biogenic amines and the lowest fat oxidation index. In addition, the peptides isolated from this variant of sausage showed the highest antioxidant potential. The obtained results justify the use of acid whey in combination with sodium ascorbate in fermented meat products without nitrogen compounds.

**Author Contributions:** Conceptualization, M.K. and D.M.S.; methodology, M.K. and A.K.; investigation, M.K. and A.K.; data curation, A.K.; writing—original draft preparation, M.K. and A.K.; writing—review and editing, D.M.S.; supervision. All authors have read and agreed to the published version of the manuscript.

**Funding:** This research was funded by the Ministry of Agriculture and Rural Development project [HOR.re.027.7.2017].

**Institutional Review Board Statement:** Not applicable.

**Informed Consent Statement:** Not applicable.

**Data Availability Statement:** Not applicable.

**Conflicts of Interest:** The authors declare no conflict of interest.

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
