# Peer review of "Effect of Acid Whey in Combination with Sodium Ascorbate on Selected Parameters Related to Proteolysis in Uncured Dry-Fermented Sausages"

_applsci, doi:10.3390/app12168316_

Round 1

Reviewer 1 Report

GENERAL COMMENTS

Dear authors,

I have some major questions.

Kind regards,

SPECIFIC COMMENTS

Any mean value presented throughout the manuscript should be presented with ± SD.

Line 89: “Jasiolka”

Dry-fermented sausage production: How many kg were produced of each of the four formulations? Before the sausages were placed in the chamber how much did each one weigh on average? The temperature and humidity to which the sausages were subjected were measured how? The weight loss of all the sausages was exactly 30%? How did they make this determination and on how many sausages?

Lines 166 and 167: I think it is important to mention whether the extraction was done in duplicate, triplicate…

Line 171:  Elimination of outliers?

Rewrite - using StatisticaTM v.13.3 software (1984–Year?) from Statsoft (StatSoft Inc., Tulsa, OK, USA).

Post-hoc comparation were identified using Tukey’s honest significant difference (HSD) test.

Line 187-189: 5.97 and 6.03? Aren't these the values indicated in the table….

“the pH does not differ significantly (day 0)”? 6.04 is significantly different from 5.98 determined for treatments S and C, respectively….

“During the manufacturing process no significant differences between the samples C and S were detected”? I do not understand...at all sampling times there were significant differences between C and S..... I think the whole paragraph on pH will have to be revised!

 Line 212: in this line it is write p < 0.05, but in line 179 wrote p ≤ 0.05?

Line 211-213: I think it would be clearer if they wrote in capital and lowercase letters.

Line 220: Rewrite this line.

 Table 3: Put the parameters in the centre of the line as in table 2, it looks better.

 Lines 234-236: And why in the TFinal (day 21) this situation does not occur?

Line 240: letters a-d? It will not be a-b?

Table 6: In tables 2 and 3 the significant differences are presented in ascending order (a,b,c,d). Why does table 6 appear in descending order (c,b,a)?

I think the letters a-c in the peptide content are misplaced…. Because, for example, 13.49 is higher than 11.05.

Line 272: 11.70 ± 0.90 mg g-1? And the value 13.49 ± 0.75 mg g-1?????

Line 284: letters a-d? It will not be a-c?

Lines 287 and 296: Table 7? Variant? Why do they write treatment in tables 2 and 3 and Variant here?

Line 296: Why Histamine has not been determined? Why haven't at least all the amines with a vasoactive effect been determined? Spermidine and agmatine have not been determined? What are the detection limits of each amine? I think the results should be presented in mg/kg as they are more representative. There are many studies in which tryptamine was the amine that showed the highest concentration and the relationship between tyramine and tryptamine is known, especially at low pH and in the presence of LAB.

Lines 383-400: Do the pH values obtained in this study favour decarboxylation?

Lines 398-400: Histamine has been determined??? Isn't that information in the materials and methods?

I think it would have been extremely important to make sensory analysis of the end sausages, which would have given another robustness to the work developed.

Author Response

REVIEWER 1: Review Report (Round 1)

The authors thank for the valuable comments of the Reviewer, which allowed for the improvement of the article. Changes to the notes have been made to the text in BLUE.

SPECIFIC COMMENTS

Any mean value presented throughout the manuscript should be presented with ± SD.

Response: All the mean results presented in the article are presented with the value of SD (means ± SD)

Line 89: “Jasiolka”

Dry-fermented sausage productionHow many kg were produced of each of the four formulations? Before the sausages were placed in the chamber how much did each one weigh on average? The temperature and humidity to which the sausages were subjected were measured how? The weight loss of all the sausages was exactly 30%? How did they make this determination and on how many sausages?

Response: The part Materials and methods has been completed according to Reviewer’s recommendation (lines 109-116)

Lines 166 and 167: I think it is important to mention whether the extraction was done in duplicate, triplicate…

Response: The part 2.6 has been completed according to Reviewer’s recommendation (lines 157-158)

Line 171:  Elimination of outliers?

Rewrite - using StatisticaTM v.13.3 software (1984–Year?) from Statsoft (StatSoft Inc., Tulsa, OK, USA).

Post-hoc comparation were identified using Tukey’s honest significant difference (HSD) test.

Response: The sentence has been rewritten according to Reviewer’s recommendation (lines 179-181)

Line 187-189: 5.97 and 6.03? Aren't these the values indicated in the table….

“the pH does not differ significantly (day 0)”? 6.04 is significantly different from 5.98 determined for treatments S and C, respectively….

“During the manufacturing process no significant differences between the samples C and S were detected”? I do not understand...at all sampling times there were significant differences between C and S..... I think the whole paragraph on pH will have to be revised!

Response: I sincerely apologize for mistakes in describing these results. The description of pH has been verified and corrected (lines 196-201).

 Line 212: in this line it is write p < 0.05, but in line 179 wrote p ≤ 0.05?

Response: I sincerely apologize for mistakes. Markings have been corrected throughout the text.

Line 211-213: I think it would be clearer if they wrote in capital and lowercase letters.

Response: according to Reviewer’s recommendation the changes have been provided throughout the text.

Line 220: Rewrite this line.

Response: The line has been rewritten according to Reviewer’s recommendation (line 231-232).

 Table 3: Put the parameters in the centre of the line as in table 2, it looks better.

Response: the correction has been made according to Reviewer’s recommendation.

 Lines 234-236: And why in the TFinal (day 21) this situation does not occur?

Response: the discussion in this matter has been completed (lines 414-424).

Line 240: letters a-d? It will not be a-b?

Response: the correction has been made according to Reviewer’s recommendation.

Table 6: In tables 2 and 3 the significant differences are presented in ascending order (a,b,c,d). Why does table 6 appear in descending order (c,b,a)?

I think the letters a-c in the peptide content are misplaced…. Because, for example, 13.49 is higher than 11.05.

Response: the correction has been made according to Reviewer’s recommendation.

Line 272: 11.70 ± 0.90 mg g-1? And the value 13.49 ± 0.75 mg g-1?????

Response: the authors meant mean values, the text has been corrected.

Line 284: letters a-d? It will not be a-c?

Response: the correction has been made according to Reviewer’s recommendation.

Lines 287 and 296: Table 7? Variant? Why do they write treatment in tables 2 and 3 and Variant here?

Response: the correction has been made according to Reviewer’s recommendation.

Line 296: Why Histamine has not been determined? Why haven't at least all the amines with a vasoactive effect been determined? Spermidine and agmatine have not been determined? What are the detection limits of each amine? I think the results should be presented in mg/kg as they are more representative. There are many studies in which tryptamine was the amine that showed the highest concentration and the relationship between tyramine and tryptamine is known, especially at low pH and in the presence of LAB.

Response: Content of the BAs was measured with a reference to the amine standards (histamine, tyramine, putrescine, cadaverine, spermidine, agmatine, spermine). Histamine, spermidine and agmatine were not detected. The text has been completed with missing information. The results of biogenic amines presented in mg/kg.

Lines 383-400: Do the pH values obtained in this study favour decarboxylation?

Response: The discussion on biogenic amines has been added (lines 445-454). The effect of pH on decarboxylation has been explained.

Lines 398-400: Histamine has been determined??? Isn't that information in the materials and methods?

Response: Content of the histamine was measured. Histamine was not detected. The text has been completed with missing information.

I think it would have been extremely important to make sensory analysis of the end sausages, which would have given another robustness to the work developed.

Response: Thank you very much for this remark. Unfortunately, we did not perform a sensory analysis. We will use this remark to improve further experiences.

Reviewer 2 Report

The manuscript entitled 'Effect of acid whey in combination with sodium ascorbate on selected parameters related to proteolysis in uncured dry-fermented sausages' measured the application of acid whey and sodium ascorbate in uncured dry-fermented sausages. Overall, the content of the manuscript is clear and well-written. Considering as above, the present version can be improved as the following points:

1. In the Dry-fermented sausage production, what is the ratio of lean and fat in the  sausage.

2. In the part of Statistical analysis, the number of replications should be indicated.

3. In line 188, “At the beginning of the experiment (day 0), the pH does not differ significantly . However, as shown in your Table 1, the pH in the day 0 showed to be different in this four groups, please have a check for this point.

4. In line 234-236, the higher content of total heme pigments and heme iron content in SAWA than C and S sample was only displayed in day 14. Then, the heme pigments and heme iron content in SAWA were decreased in day 21, which had the different trend in C, S, and SAW. The reason on this matter should be discussed.

5. In line 367-369, the antioxidant activity of peptides in SAWA showed to be higher compared to C and S. The reason on this matter should be illustrated. Is there some differences on peptides characteristics(molecule weight, peptides chain,hydrophobic  properties)in SAWA.

6. According to your results, the application of acid whey and sodium ascorbate significantly reduced the content of biogenic amines in fermented sausages. The major mechanism on this reducing effect on biogenic amines was not discussed, please illustrate this point.

Author Response

REVIEWER 2: Review Report (Round 1)

The authors thank for the valuable comments of the Reviewer, which allowed for the improvement of the article. Changes to the notes have been made to the text in red.

The manuscript entitled 'Effect of acid whey in combination with sodium ascorbate on selected parameters related to proteolysis in uncured dry-fermented sausages' measured the application of acid whey and sodium ascorbate in uncured dry-fermented sausages. Overall, the content of the manuscript is clear and well-written. Considering as above, the present version can be improved as the following points:

  1. In the Dry-fermented sausage production, what is the ratio of lean and fat in the sausage.

Response: Information “Formulation of dry fermented sausage assumed the use of meat and tallow in the ratio of 80:20”has been included in the Materials and methods section.

  1. In the part of Statistical analysis, the number of replications should be indicated.

Response: Information “The experiment was replicated three times by producing three different batches on separate days” has been included in the Statistical analysis section.

  1. In line 188, “At the beginning of the experiment (day 0), the pH does not differ significantly ”. However, as shown in your Table 1, the pH in the day 0 showed to be different in this four groups, please have a check for this point.

Response: I sincerely apologize for mistakes in describing these results. The description of pH has been verified and corrected.

  1. In line 234-236, the higher content of total heme pigments and heme iron content in SAWA than C and S sample was only displayed in day 14. Then, the heme pigments and heme iron content in SAWA were decreased in day 21, which had the different trend in C, S, and SAW. The reason on this matter should be discussed.

Response: The discussion on the total heme pigments and heme iron has been completed.

  1. In line 367-369, the antioxidant activity of peptides in SAWA showed to be higher compared to C and S. The reason on this matter should be illustrated. Is there some differences on peptides characteristics(molecule weight, peptides chain,hydrophobic properties)in SAWA.

Response: The discussion on the antioxidant properties of peptides has been completed.

  1. According to your results, the application of acid whey and sodium ascorbate significantly reduced the content of biogenic amines in fermented sausages. The major mechanism on this reducing effect on biogenic amines was not discussed, please illustrate this point.

Response: The discussion on biogenic amines in the indicated range has been added.

Round 2

Reviewer 1 Report

GENERAL COMMENTS

Dear authors,

Congratulations on the manuscript,

I have some minor questions.

Kind regards,

Reviewer

SPECIFIC COMMENTS

Dear’s authors, any mean value presented throughout the manuscript should be presented with ± SD. For example, in line 284, when you write 9.69 to 11.70 mg g-1of product. You should write: 9.69 ± 1.19 to 11.70 ± 0.49 mg g-1. You should write this way throughout the manuscript.

Line 100: insert a space between Poland)were.

Lines 115 and 116: The weight loss was verified as? With a scale? Brand, model, city and country of production?

Line 176:  insert a space between wasexpressed. mg g-1? In line 307 you wrote mg kg-1….

 Line 177:  I stand by my question, have the outliers been determined, and how?

Line 196-198: Dear authors, I don't understand! I stand by my question; I don't see the values 5.97 and 6.03 in Table 2!

 Line 199: P ≥ 0.05???? Will be P ≤ 0.05?

 Lines 224 and 226: in line 224 the authors write P, but in 226 they write P. Please, standardise.

 Lines 301 and 302: You must write down what the detection limits of the method were for the not detected amines.

Author Response

The authors thank the Reviewer for the valuable comments, which allowed for the improvement of the article. Changes to the notes have been made to the text in GREEN.

Dear’s authors, any mean value presented throughout the manuscript should be presented with ± SD. For example, in line 284, when you write 9.69 to 11.70 mg g-1of product. You should write: 9.69 ± 1.19 to 11.70 ± 0.49 mg g-1. You should write this way throughout the manuscript.

Response: I sincerely apologize for misunderstanding. I have completed the article as recommended.

Line 100: insert a space between Poland)were.

Response: I inserted a space.

Lines 115 and 116: The weight loss was verified as? With a scale? Brand, model, city and country of production?

Response: I supplemented this part of the work with information “with the PS1000.X2 scale (RAWAG, Radom, Poland)”.

Line 176:  insert a space between wasexpressed. mg g-1? In line 307 you wrote mg kg-1….

Response: I inserted a space. I changed records on “The BAs concentration was expressed as mg kg-1 of dry fermented sausages”.

 Line 177:  I stand by my question, have the outliers been determined, and how?

Response: I supplemented this part of the work with information “The outliers have been determined using the interquartile range”.

Line 196-198: Dear authors, I don't understand! I stand by my question; I don't see the values 5.97 and 6.03 in Table 2!

Response: I sincerely apologize for mistake. I have changed the data.

 Line 199: P ≥ 0.05???? Will be P ≤ 0.05?

Response: I sincerely apologize for mistakes. I changed “P ≥ 0.05 to P ≤ 0.05”.

 Lines 224 and 226: in line 224 the authors write P, but in 226 they write P. Please, standardise.

Response: I sincerely apologize for mistakes. The record has been standardised.

 Lines 301 and 302: You must write down what the detection limits of the method were for the not detected amines.

Response: I supplemented this part of the work with information “The detection limits of the method were 100 pmol for histamine and agmantine and 50 pmol for spermidine”.